# Serotonin Signalling in Flatworms: An Immunocytochemical Localisation of 5-HT_7_ Type of Serotonin Receptors in *Opisthorchis felineus* and *Hymenolepis diminuta*

**DOI:** 10.3390/biom11081212

**Published:** 2021-08-15

**Authors:** Natalia Kreshchenko, Nadezhda Terenina, Artem Ermakov

**Affiliations:** 1Institute of Cell Biophysics of Russian Academy of Sciences, 142290 Pushchino, Russia; 2Center of Parasitology A.N. Severtsov Institute of Ecology and Evolution of Russian Academy of Sciences, 119071 Moscow, Russia; terenina_n@mail.ru; 3Institute of Theoretical and Experimental Biophysics Russian Academy of Sciences, 142290 Pushchino, Russia; beoluchi@yandex.ru

**Keywords:** serotonin, serotonin receptors, flatworms, immunocytochemistry, confocal laser scanning microscopy

## Abstract

The study is dedicated to the investigation of serotonin (5-hydroxytryptamine, 5-HT) and 5-HT_7_ type serotonin receptor of localisation in larvae of two parasitic flatworms *Opisthorchis felineus* (Rivolta, 1884) Blanchard, 1895 and *Hymenolepis diminuta* Rudolphi, 1819, performed using the immunocytochemical method and confocal laser scanning microscopy (CLSM). Using whole mount preparations and specific antibodies, a microscopic analysis of the spatial distribution of 5-HT_7_-immunoreactivity(-IR) was revealed in worm tissue. In metacercariae of *O. felineus* 5-HT_7_-IR was observed in the main nerve cords and in the head commissure connecting the head ganglia. The presence of 5-HT_7_-IR was also found in several structures located on the oral sucker. 5-HT_7_-IR was evident in the round glandular cells scattered throughout the larva body. In cysticercoids of *H. diminuta* immunostaining to 5-HT_7_ was found in flame cells of the excretory system. Weak staining to 5-HT_7_ was observed along the longitudinal and transverse muscle fibres comprising the body wall and musculature of suckers, in thin longitudinal nerve cords and a connective commissure of the central nervous system. Available publications on serotonin action in flatworms and serotonin receptors identification were reviewed. Own results and the published data indicate that the muscular structures of flatworms are deeply supplied by 5-HT_7_-IR elements. It suggests that the 5-HT_7_ type receptor can mediate the serotonin action in the investigated species and is an important component of the flatworm motor control system. The study of the neurochemical basis of parasitic flatworms can play an important role in the solution of fundamental problems in early development of the nervous system and the evolution of neuronal signalling components.

## 1. Introduction

Serotonin is a biogenic amine, widely distributed in the plant and animal kingdoms. It was found in coelenterates, molluscs, crustaceans and other invertebrates. Serotonin is synthesised from tryptophan amino acid by hydroxylation and decarboxylation. In vertebrates, serotonin causes the smooth muscle contraction of the intestine, uterus, bronchus and blood vessels, the latter leading to the vasoconstriction [1,2,3,4]. In higher animals and in humans, serotonin’s biological roles include sleep regulation and support of psycho-emotional reactions: fear, anxiety and disturbance. Serotonin level increases in the state of euphoria, that is why it has been named the ‘hormone of happiness’. Serotonin also plays a role in the regulation of circadian rhythms, hormones secretion, feeding and sexual behaviour, as well as immune response and metabolism [5,6,7,8,9,10,11]. In a broad group of invertebrate animals, the information on serotonin localisation and function remains limited.

Parasitic flatworms (belonging to the phylum Platyhelminthes) are a diverse and widespread group of organisms. They have tremendous medical, agricultural and economic value. They develop with a complicated life cycle which may include several intermediate hosts and free living stages. A few distinctive experimental models exist among parasitic flatworms (for example, *Schistosoma mansoni, Mesocestoides vogae* and *Hymenolepis diminuta*) [12,13,14]. Whole genome sequence has been obtained for blood parasite trematodes *S. mansoni* and *S. japonica* [15,16,17]. For several other flatworm species, such as *Schmidtea mediterranea*, *Clonorchis sinensis, S. haematobium, Haemonchus contortus, Echinococcus multilocularis, E. granulosus, Taenia solium* and the laboratory model *Hymenolepis microstoma* genome sequences were also recently annotated [18,19,20,21,22].

The flatworm’s nervous system plays a central role in the realisation of organism vital functions: such as locomotion, feeding, migration, host seeking and reproduction. In spite of a significant research interest in some morpho-functional aspects of these organisms, particularly the neuromediators and their functional value remain poorly studied. The nervous system of parasitic flatworms contains a variety of signalling molecules, including serotonin which has been found in all flatworms investigated so far [23,24,25]. It has been shown by the immunocytochemical method that the localisation of this substance is associated with the central and peripheral parts of the nervous system of helminths [26,27,28,29,30,31,32]. The presence of serotonin in trematodes and cestodes has also been shown by biochemical methods [32,33,34,35,36,37].

Available data indicate that larval and adult forms of parasitic flatworms—cestodes and trematodes, are capable of active absorption of serotonin from their host through a highly specialised serotonin-transport system [33,38,39,40,41,42]. At the same time, it has been found that parasitic flatworms can synthesise serotonin using the serotonin synthesis enzyme, tryptophan hydroxylase, detected in various representatives of flatworm, expressing in both the parasitic and free-living stages of flatworm development. Thus, serotonin transporters and the synthesis pathway were identified in *S. mansoni* [43,44,45,46] and in *H. diminuta* [37,47,48,49].

The functions of serotonin in Platyhelminthes remain poorly studied. However, the regulation of motor activity of parasitic worms is a major identified serotonin function. Several serotonin effects have been documented: the stimulation of the larval motility of the cestode *Mesocestoides corti* [50], free-swimming larvae (cercariae) of trematodes *Cryptocotyle lingua* and *Himasthla elongata* [51], sporocysts of *S. mansoni* [12], locomotion of *Haplometra cylindracea* [52]. Serotonin induced an excitation of muscle strip contractions in *H. diminuta* [53], *Fasciola hepatica* [54,55] and *Diclodophora merlangi* [56]. Moreover, serotonin was found to be prerequisite for musculature functioning in *S. mansoni* [12,57,58,59,60]. The stimulation of body surface cilia beating [61], whole worm twisting and behavioural changes were observed in free-living flatworms (planarians) under serotonin administration [62]. Serotonin inducing contractions of the isolated muscle fibres in planarian *Procerodes littoralis* was further verified [63]. Serotonin also stimulated the development of metacestodes of *E. multilocularis* [64], regulating RNA synthesis in flatworms [65] and was essential for eye regeneration in planarians *S. mediterranea* [66] and *Girardia tigrina* [67], as well as accelerated growth of planarian regenerative blastema in [68].(Kreshchenko, personal report).

A physiological action of serotonin is carried out by the activation of specific serotonin receptors. Up to 14 receptor subtypes for serotonin have already been found in vertebrates grouped into seven families (5-HT_1_-5-HT_7_), six of which (5-HT_1_, 5-HT_2_, 5-HT_4_–5-HT_7_) are G-protein coupled receptors (GPCRs) and one, 5-HT_3_ includes ionotropic receptors [69,70,71,72,73]. G-protein coupled receptors represent the largest known group of membrane proteins extending throughout the Metazoan.

Among invertebrates, the serotonin and its receptors had been identified in nematodes [74,75,76,77], insects [78,79], molluscs [80], flatworms [24,61,81,82]. They found to be difficult to classify because they do not always recognise classical (i.e., mammalian) serotonin agonists or antagonists [83]. In invertebrates the information was limited to three identified serotonin receptors families (5-HT_1_, 5-HT_2_ and 5-HT_7_), known to be orthologous to mammalian ones [24,78,83]. Recently, the presence of five serotonin receptor families, namely 5-HT_1_, 5-HT_2_, 5-HT_4_, 5-HT_6_ and 5-HT_7_ have been reported among the invertebrates: arthropods, annelids, nematodes and molluscs [84,85].

The aim of this study was to identify the components of the serotonin signalling system including the serotonin and serotonin receptors of 5-HT_7_ subtype in the larvae of cestode *Hymenolepis diminuta* Rudolphl 1819, Blanchard 1891 and larvae of trematode *Opisthorchis felineus* Rivolta 1884, by means of immunocytochemistry, fluorescent microscopy and confocal laser scanning microscopy. Trematode *O. felineus* causes a dangerous disease in livestock animals and human beings, opisthorchiasis, which is a broadly distributed from East Europe to Central Asia. Hymenolepiasis is a parasitic disease caused by cestode *H. diminuta*, which is zoonotic flatworm specie widespread worldwide.

The data obtained in the present study on the serotonin signalling in flatworms expand our knowledge on the function of the flatworm nervous system. The results can have significance for the development of a new pharmacological strategy against helminth infections by affecting the functioning of the nervous system of parasitic flatworms and neuronal signal compounds.

## 2. Materials and Methods

In this study, the cestode larvae (cysticercoids) of *H. diminuta* (Hymenolepididae) and trematode larvae (metacercariae) of *O. felineus* (Opisthorchiidae) have been used. The presence and localization of serotonin and serotonin receptor 5-HT_7_ types in whole mounts of cestode and trematode larvae tissues was studied by indirect immunocytochemical method [86].

### 2.1. Sampling and Fixation

Metacercariae of *O. felineus* were obtained from naturally infected fish *Leuciscus idus* captured from water reservoir in Tobolsk region, Russia. Fish muscles were digested in artificial gastric juice at 37 °C [87] and collected metacercariae were excysted in 3–5 drops of 0.25% trypsin (St. Louis, Sigma, MO, USA) in 10 mL of saline solution (0.85% *w*/*v* sodium chloride in distilled water) at 37 °C during 10–20 min.

Cysticercoids of *H. diminuta* were obtained from the abdominal body cavity of the experimentally infected flour beetle *Tenebrio molitor* (Insect, Tenebrionidae). Cysticercoids were excysted according to a modified Rothman’s procedure [88]. Cestode larvae were treated with 1% pepsin (St. Louis, Sigma, MO, USA) and 1% HCl for 30 min in Ringer’s solution (Obninsk, Hemofarm, Russia) at room temperature (RT). Then, cysticercoids were transferred to 0.5% trypsin (St. Louis, Sigma, MO, USA) and 0.1% sodium taurocholate for 20–25 min at 37 °C and then rinsed again in Ringer solution.

The worms were flat fixed in 4% paraformaldehyde (Santa Ana, MP Biomedicals, CA, USA) in 0.1 M phosphate buffered saline with pH 7.4 (PBS, St. Louis, Sigma, MO, USA) at 4 °C for 12 h and then transferred for storage in PBS with 10% sucrose (Sigma, USA) and kept for 5–7 days at 4 °C (until stained).

### 2.2. Immunocytochemistry

Samples were washed for 12 h in PBST solution containing 0.1 M PBS, 0.3% Triton X-100 (Sigma, Burlington, MA, USA) and 0.1% bovine serum albumin (Radnor, Amresco, PA, USA) at 4 °C. After that they were incubated for five days in polyclonal primary rabbit antiserum to serotonin (1:500, Hudson, Immunostar, WI, USA, Product ID: 20080, RRID: AB_572263) or with 5-HT (Serotonin) 7 receptor polyclonal primary antibodies (1:300, Hudson, Immunostar, WI, USA, Product ID: 24430, RRID: AB_572214) diluted in 0.1 M PBS at 4 °C. After a wash in PBS for 6 h at 4 °C, the samples were incubated with the secondary fluorescently-labeled AlexaFluor488 immunoglobulins (goat anti-rabbit IgG (H+L), Waltham, Abcam, MA, USA, Cat# ab150077, RRID: AB_2630356) diluted (1:400) in 0.1 M PBS for next five days at 4 °C. Controls included: (1) incubation of samples with only secondary immunoglobulins without primary antibodies and (2) using of non-immune rabbit serum instead of the primary antiserum. The negative controls demonstrated the absence of specific staining in the worm’s tissue.

Due to a lack of full genomes in species studied (*O. felineus* and *H. diminuta*) the molecular characterization (sequencing and cloning) of the serotonin receptors was not performed (and was not a task of the study). As a consequence, the receptor proteins are also not identified. It will be of great interest for us (or other researchers) to perform such a study in the future. That’s why the precise control of the staining specificity was not possible. The antibodies reported were raised against the conservative sequence of the the 5-HT_7_ vertebrate receptor. Assuming that flatworm serotonin receptor can be the most ancient one, it should contain this conservative motif in the investigated species. Nevertheless, our ICC results on the 5-HT_7_ serotonin receptors localisation in *O. felineus* and *H. diminuta* should be considered as a preliminary, meaning the 5-HT_7_-like-immunoreactivity. Only bioinformatic and comparative approaches were used to support our findings.

### 2.3. Histochemical Staining

For identification of actin of muscle filaments the histochemical staining of the samples by TRITC(tetramethylrhodamine B isothiocyanate)-conjugated phalloidin (St. Louis, Sigma, MO, USA, Cat# P1951) was performed in dilution of 1:200 in PBS for 18-24 hrs. All procedures were performed in dark at 4 °C.

### 2.4. Confocal Laser Scanning and Fluorescent Microscopies

For microscopic analysis 7–10 replicas of each species have been used. Specimens were analysed under the fluorescent microscope Leica DM6000B equipped with a digital camera DC300F (Wetzlar, Leica Microsystems, Germany) and confocal laser scanning microscope Leica TCS SP5 (Wetzlar, Leica Microsystems, Germany) at the Optical Microscopy and Spectrophotometry Core Facilities of Federal Research Center “Pushchino Scientific Center for Biological Research of the Russian Academy of Sciences” (Moscow Region, Russia). For Leica DM6000B microscopic analysis a fluorescent filter I3 (excitation spectrum of 450–490 nm, emission spectrum of 515 nm) was used for the identification of Alexa488 fluorophore, while TRITC fluorophore was detected with a N2.1 filter (excitation spectrum of 515–560 nm; emission spectrum of 590 nm). For CLSM Leica TCS SP5 analysis, the microphotographs presented are either a maximal projection of a total of 16 to 32 consequent optical sections reconstructed at maximum fluorescence intensity, or a single optical section (or a snapshot), obtained by scanning through 30–40 μm sample thickness.

### 2.5. Bioinformatics

A phylogenetic analysis of 5-HT_7_ receptor was performed for several flatworm species with fully annotated genome sequences. A comparative analysis of serotonin receptor 5-HT_7_ genes in cestodes *H. diminuta*, *H. microstoma*, *M. corti*, *E. granulosus*, *Taenia asiatica*, trematodes *F. hepatica*, *Opisthorchis viverrini*, *Schistosoma japonicum*, *S. mansoni* and planarians *S. mediterranea, D. japonica* and mammals *H. sapiens* and *M. musculus* was performed using NCBI database. The phylogenic tree (Section 3.3) was automatically constructed with Unipro UGENE software (version 39.0) using NCBI Protein database [89]. Since the genomes of *O. felineus* and *H. diminuta* are partially annotated and of poor quality, we used 5-HT_7_ amino acid sequences of closely related species to construct the tree, where the annotation was performed with a higher quality.

However, despite of the absence of full genomes, we also performed the bioinformatic search and analysis of the 5-HT_7_ serotonin receptor gene sequences between the partial genomes (annotated genes sequences) for *H. diminuta* and *O. felineus*. The amino acid sequence of the gene for the serotonin receptor 5-HTR-7 (5-hydroxytryptamine receptor 7) *Mus musculus* (NP_032341.2) was used for the search. Search for homologues of serotonin 5-HT_7_ receptors in genomes of *O. felineus* and *H. diminuta* was carried out using the WormBase Parasite database [90] which contains the partly annotated genomes and gene sequences and the NCBI genome database [91]. The search was performed using blastp in modification for distant homology. The analysis show that the closest orthologue of this receptor in *H. diminuta* possesses of a sequence located in N-terminal region, showing 43% similarity with the receptor 5-HT_7_ protein of *Mus musculus*. Similarly, the nearest 5-HT_7_ ortholog in *O. felineus*, has also a 38% similarity to the peptide in the N-terminal region. In the available genome sequences for *O. felineus* and *H. diminuta*, the presence of 1, 5 and 7 types of serotonin receptors characteristic of higher animals were also reliably determined.

## 3. Results

### 3.1. Opisthorchis felineus Metacercariae: Immuoreactivity to Serotonin and 5-HT_7_ Serotonin Receptor

Immunoreactivity to serotonin (5-HT-IR) is identified in the central and peripheral departments of nervous system of *O. felineus* metacercariae: in the brain ganglia and the connecting commissure, in major nerve cords, in several pairs of serotonin neurons located along the main nerve cords (Figure 1a–c), as well as in the 5-HT-IR nerve fibres running towards the oral and ventral sucker (Figure 1b,c inset).

Strong immunostaining to serotonin receptor (5-HT_7_-IR) has been detected in the compartments of the central nervous system: in the cerebral ganglia, commissure connecting the brain ganglia, in major (ventral) nerve cords (Figure 1d–g) and in the nerve cells along the ventral cords (Figure 1e–g,i). Weaker 5-HT_7_-IR is found in nerve branches running from the brain ganglia towards the oral sucker (Figure 1e).

A rather intense immunostaining to 5-HT_7_ has also been observed in numerous small round or oval structures (about 3–4 μm of size) scattered throughout the worm body and localised both deep in the body and near the body surface (Figure 1h, inset,i,k,l). These cells are, presumably, glandular cells or the elements of the excretory system (flame cells), or other unidentified cells of the parasite body (Figure 1k). A row of such 5-HT_7_-IR structures has been observed along the worm body margins (Figure 1i,l).

Staining to 5-HT_7_ was visible in structures, located in the oral sucker and corresponding probably to the sensory papillae (Figure 1j, inset). In the ventral sucker which is richly supplied by serotonergic nervous components (Figure 1b, inset), thin 5-HT_7_-IR nerve fibres were detected (Figure 1d, inset). The positive staining to 5-HT_7_ was also prominent in two serotonergic neurons located near the pharynx (Figure 1j,k) and along the nerve fibres near the esophagus (Figure 1k).

When the muscles of the metacercariae of *O. felineus* were stained with phalloidin, the circular, longitudinal and diagonal fibres of the body wall, as well as the fibres running from the ventral sucker to the body wall were clearly visible (Figure 2a–c). The 5-HT_7_-IR elements looking like small dots were regularly distributed along the longitudinal, circular and diagonal muscle fibres of the body wall (Figure 2d,e,g–i). The staining was also evident along single muscle fibres extending from the body wall to the ventral sucker (Figure 2f,j,k). We observed also 5-HT_7_-IR in the longitudinal muscle fibres of the intestine (Figure 2f) and the excretory bladder (Figure 2k). Punctate pattern of 5-HT_7_-IR observed along the muscle fibres of the body indicates that the receptor is localised either on the nerve fibres that are closely associated with the musculature, or at the muscle fibres itself.

### 3.2. Hymenolepis diminuta Cysticercoids, Immunoreactivity to Serotonin and 5-HT_7_ Serotonin Receptor

The localisation and distribution of serotonin immunoreactivity (5-HT-IR) was observed in cell bodies and nerve fibres of the central nervous system of cysticercoid larvae of *H. diminuta* (Figure 3a–f). Serotonergic nerve elements were present in the lateral or “cerebral” ganglia, in the connective commissure, in the rostellar ganglia, as well as in the longitudinal nerve cords and in transversal connective commissures (Figure 3b,c,e). A pair of serotonergic nerves (the rostellar nerves) was found running from the lateral ganglia to the rostellar ganglia to join them (Figure 3b). Several 5-HT-IR neurons (sizes 4.6–7.4 µm, *n* = 5, where n is a number of specimens measured) have been identified in each of two lateral ganglia. From 14 to 18 serotonergic neurons (*n* = 4) can be observed in the scolex. 5-HT-IR nerve fibres are present in the nerve plexus within each sucker (Figure 3c,f). Positive staining to serotonin has been observed in the neurites comprising the longitudinal nerve cords, two of the cords are the most pronounced (Figure 3b,d,e) with the distance between them measuring about 50.6–64.8 µm (*n* = 5). Serotonergic neurons (size from 4.4 to 6.9 µm, *n* = 4) were detected along the major nerve cords of the worm body (Figure 3b–e). Several thin 5-HT-IR transversal nerve commissures are connecting the longitudinal nerve cords (Figure 3b,d,e). In the posterior region of the cysticercoid a rich network of serotonergic neuritis is localised (Figure 3f).

A weak staining to the serotonin receptor 5-HT_7_ was found in *H. diminuta* cysticercoids, namely in thin longitudinal nerve cords. The 5-HT_7_-IR was detected along the longitudinal and transversal muscle fibres of the body wall musculature and in the muscle fibres of suckers (Figure 3g–j). Strong 5HT_7_ immunoreactivity was observed in the oval structures (with the length of 4.2–5.4 μm and the width of 2.4–2.8 μm, *n* = 7), probably the flame cells of the excretory system and/or other unidentified cells, scattered throughout the larvae body (Figure 3g–k,m), which were also stained with phalloidin (Figure 3l). The heterogenic staining to 5-HT_7_ was noted in these structures.

### 3.3. A phylogenetic Analysis of 5-HT_7_ Receptor

A phylogenetic analysis of 5-HT_7_ receptor was performed for several flatworm species with fully annotated genome sequences. A comparative analysis of serotonin receptor 5-HT_7_ genes in cestodes *H. diminuta*, *H. microstoma*, *M. corti*, *E. granulosus, Taenia asiatica*, trematodes *F. hepatica*, *Opisthorchis viverrini, Schistosoma japonicum, S. mansoni* and planarians *S. mediterranea, D. japonica* and mammals *H. sapiens* and *M. musculus* was performed using NCBI database. The phylogenic tree (Figure 4) was constructed. The presence in the genome of investigated species the serotonin 5-HT_7_ receptor was revealed. The branch support analysis indicated that genes of 5-HT_7_ cestode receptor are clustered together and separated from the trematode ones. Planarian 5-HT_7_ receptor has branched from the common ancestor with other trematode species. Serotonin 5-HT_7_ receptor of cestodes *O. viverrini* and *T. asiatica* (Figure 4, Appendix A) branched off earlier from the receptors of other cestode and trematode species. The data also indicate that the cestode 5-HT_7_ serotonin receptor is the most ancient among the flatworms.

## 4. Discussion

The presence of serotonin in adult and larval stages of parasitic flatworms—trematodes, cestodes and monogeneans was shown in a large number of species [27,28,32,37,92,93,94,95,96,97,98]. Serotonin activate specific serotonin receptors, six of which belong to the rhodopsin subfamily of GPCRs, consisting of seven transmembrane domains and 5-HT_3_—a serotonin-gated ion channel [73,84,99]. Serotonin receptors in Platyhelminthes have been demonstrating in several studies [59,60,100]. Using the pharmacological approach, the presence of serotonin receptors in free-living planarians *Polycelis tenuis*, *Dugesia gonocephala* and *D. lugubris* was shown [81,100,101]. The investigations suggested that cAMP mediates the physiological action of serotonin in planarians [101], but the receptors themselves were not identified at the molecular level at that time. In planarian *D. japonica*, four serotonin G-protein coupled receptors were found, the nucleotide sequences of which were established and have significant homology with the 5-HT_1A_ human serotonin receptor and Drosophila 5-HTdrol receptor. The planarian serotonin receptors were named 5-HTLpla1-4 [102,103,104]. Another receptor (DjSER7) was isolated from the planarian *D. japonica* demonstrating a high affinity for serotonin when incorporated into *Xenopus* oocytes [105].

The pharmacology of serotonin receptors has also been investigated in the in liver fluke *F. hepatica* [106,107,108], in the human parasite *S. mansoni* [60,109,110] and in cestodes *E. granulosus* and *M. corti* [82]. In the study of Chan et al. [110], a large battery of pharmacological compounds was screened to compare the drug activity at 5-HT_7a_Sch receptors of *S. mansoni* with that of human 5-HT_7_ receptor. Most drugs displayed a higher potency with the human receptor, but some were more active with 5-HT_7a_Sch.

Using a bioinformatics approach, in genome databases, a large number of heptaspiral receptors coupled to G-proteins has been identified, including 24 in trematode *S. mansoni* and 66 in planarian *S. mediterranea* that were considered as putative aminergic receptors [111]. These GPCRs were activated by a number of biologically active substances, some of them by serotonin [112]. However, the serotonin receptors had not been cloned or characterised at the molecular level in any of the parasitic flatworms until 2014 [60]. In the work of Patocka et al. [60], the first molecular evidence of a functional serotonin receptor (Sm5HTR) in *S. mansoni* was provided. The schistosome receptor has been found to be closely related to the type 7 (5-HT_7_) serotonin receptors. The functional significance of the serotonin receptor was demonstrated. It was shown that a decrease in the motor activity of *S. mansoni* adult worms was correlated with decreased of5-HT_7_ expression as a result of its RNAi [60]. The authors also indicated that they were unable to find the 5-HT_2_ receptors in *S. mansoni* and suggested that this type of receptor may have been lost in the organism or even in the phylum [60].

Bioinformatics genome sequence searches have predicted the presence of additional 5-HT receptors in flatworms [112,113]. Recently, three serotonin receptors were cloned, sequenced and functionally characterised in *E. granulosus* and *M. corti*. These new GPCR receptors exhibit unique characteristics, including a particular sensitivity to serotonin as well as a distinctive pharmacology [82]. A set of 147 GPCRs was also reported in *F. hepatica* [113]. Among them, 38 aminergic receptors from the α subfamily of rhodopsins were detected by in silico ligand-receptor predictions. The comparison performed with known deorphanised receptors and positional conservation of ligand-interacting residues indicated that five of the *F. hepatica* receptors bore about ≥80% identity with the human 5-HT_1A_ serotonin receptor. These amino acid residues were also conserved in the deorphanised of *S. mansoni* serotonin receptor Sm5HTR, identified by Patocka et al. [60]. Three receptors also resembled Sm5HTR in phylogenetic analysis, being likely 5-HT receptors [113]. Thus, the 5-HT_7_ receptors have been described in the members of the Platyhelminthes phylum and this receptor type appears to be the dominant clade in this group of organisms [60,82,105,110,113,114].

In the present study, we have obtained, for the first time data on the presence of 5-HT_7_ immunoreactivity in the tissues of two flatworms—trematode *O. felineus* and cestode *H. diminuta,* species widely distributed in densely populated regions of Europe and Western Siberia. To date, information on the components of the serotonergic system in *O. felineus* is restricted to a few studies where serotonin was detected immunocytochemically in tissue of adults and larvae of *O. felineus* [29,115]. Our present investigation confirms the previous data on serotonin distribution in central and peripheral nervous systems of the metacercariae of *O. felineus*. Our study show, for the first time, that 5-HT_7_ immunostaining is abundant in the tissue of trematode *O. felineus* metacercariae.

### 4.1. Serotonin and 5-HT_7_-IR in Metacercariae of O. felineus

5-HT_7_-immunopositive staining is localised in the compartments of the metacercariae central nervous system—in the brain ganglia, brain commissure connecting the ganglia, the major nerve cords and in the peripheral compartments of the nervous system. The present results may indicate that the 5-HT_7_ type of serotonin receptor can mediate the serotonin action and is a component of the parasite motor system already existing in the larval stage of *O. felineus* development.

#### 4.1.1. Ventral Sucker

We observed 5-HT_7_-IR in the fibres located among the muscle filaments of the ventral sucker of *O. felineus*, which is richly supplied by serotonergic fibres [29]. Data on the innervation of the attachment organ by serotonergic neuritis was also shown for other trematode species [116] suggesting an important role of serotonin in the regulation of the ventral sucker muscle activity. Our findings are in accordance with the results obtained on *S. mansoni* [60], proposing that serotonin released from serotonergic neurons could activate the5-HT_7_ receptor to control the musculature of the suckers and, therefore, control the worm’s ability to attach to the host organism and feed.

#### 4.1.2. Sensory Structures (Papillae)

As it is known, various types of sensory receptors exist on the body surface of trematode larvae—cercariae and metacercariae. Thus, the presence of sensory papillae on oral and ventral suckers was revealed intrematodes *Gorgoderina vitelliloba* [117], *Phyllodistomum conostomum* [118], *Glosidium pedatum* [119], *F. hepatica* [120] and *Leucochloridium sp* [121]. Sensory papillae had been found in the oral and ventral suckers of *O. viverini* metacercariae [122]. Several types of sensory structures (sensory receptors) have also been described at the body surface of the cercariae of several trematodes [123]. There is evidence that serotonin is present in the sensory nerve endings of trematodes and cestodes [37,94]. Innervation of the oral sucker by serotonergic neurites in *O. felineus* and in other trematodes was shown previously [29,115,116]. These data predict the involvement of serotonin in the function of the sensitive organs of parasites. Our study revealed that in metacercariae of *O. felineus,* 5-HT_7_-IR is exhibited in the sensory papillae of the oral sucker. The widespread distribution of 5-HT_7_-IR in *O. felineus* larvae obtained in the present work suggests that besides motor control, serotonin receptors may perform other activities. The present data on *O. felineus* metacercariae are in agreement with the results obtained for *S. mansoni,* namely, that serotonin may act through the Sm5HTR receptor and modulate the sensory circuits at the worm’s surface, representing an important mechanism of host-parasite interaction [60].

#### 4.1.3. Glandular and Excretory Cells

In *O. felineus* metacercariae we observed 5-HT_7_-IR in the numerous round structures that were scattered throughout the worm body. We suggest that these structures can be the secretory gland cells described in a number of larvae and adults of trematode species [124,125,126]. In the larvae they synthesise the material for cysts including some mucopolysaccharides. In the oral and ventral suckers, such glands release the secret for the better penetration or/and attachment of the parasite to the host tissue. There are some data of interest on other invertebrates, namely, insects. The insect 5-HT_7_ receptor was found to be highly expressed in the salivary gland and thought to be involved in the induction of saliva secretion by serotonin [127,128]. The 5-HT_7_-IR observed in the round structures of the body of *O. felineus* metacercariae can also be associated with other elements, such as the flame cells of the excretory system, which requires additional research.

#### 4.1.4. The Digestive System

In *O. felineus* metacercariae we observed strong 5-HT_7_immunopositive staining in two nerve cells, located near the pharynx and in the neurites along the oesophagus. This region is also innervated by serotonergic neuritis [29]. The results suggest that serotonin can activate the 5-HT_7_ receptor to control the musculature of the digestive system of parasite larvae. Strong Sm5HTR expression was also shown in the developing caecum of schistosomes larvae, implying a potential serotonin role in the regulation of the gut activity of the parasite [60]. It is interesting to note, that in the insect *Aedes aegypti,* two axons have been labelled with antibodies against the 5-HT_7_ receptor, running in parallel along the hindgut [129]. It is believed that 5-HT_7_ receptor-mediated regulation of the visceral muscle activity in the gastrointestinal tract can be evolutionarily conserved between invertebrates and vertebrates, because it was demonstrated that the 5-HT_7_ receptors mediated smooth muscle relaxation of the gastrointestinal tract both in mammals and in insects [78,130].

#### 4.1.5. Musculature

In metacercariae of *O. felineus* the 5-HT_7_ serotonin receptor was abundantly expressed along the muscle fibres of the body musculature, following the musculature pattern identified by phalloidin staining. Punctate staining was observed along the longitudinal, circular and diagonal muscle fibres of the body wall, as well as along the muscle filaments around the intestine, along the muscle fibres extending from the body wall to the ventral sucker and in the musculature comprising the oral and ventral suckers. This finding can indicate the important role of the serotonin signalling system in the regulation of parasite muscle activity, confirmed by the presence of serotonergic neurites in *O. felineus* musculature [29]. More research is needed to elucidate the function of oval structures in the *O. felineus* metacercariae body, staining with anti-5-HT_7_. A comparative analysis of 5-HT_7_-IR localisation in different life cycle stages of *O. felineus*—in the parasitic adult stage and free living larvae (cercariae) would also be of great interest.

The investigation of Patocka et al. [60] on *S. mansoni* has revealed that Sm5HTR is abundantly expressed in the nervous systems of both schistosomules and adult worms. A staining pattern was also observed in the body wall musculature of *S. mansoni* larvae. The study indicated that Sm5HTR is just as important for motor control in the adult worms as in the larvae. The authors demonstrated that the serotonin receptor is expressed in close proximity to the serotonin-containing nerve fibres, where it could be activated by the endogenously released transmitter. The results proposed that serotonin is acting through Sm5HTR to stimulate the release of other neurotransmitters and, besides neuromodulation, may have direct effects on the worm’s body wall muscles [60]. Generally, the pattern of 5-HT_7_-IR localisation in metacercariae of *O. felineus* identified in the present study is consistent with those obtained for *S. mansoni* [60]. It should be noted that, to date, it is the only known immunocytochemical detection of serotonin receptors in trematodes [60].

Thus, on the basis of fluorescent immunocytochemistry, the presence and localisation of serotonin and serotonin 5-HT_7_ receptor immunoreactivities were shown in the tissues of *O. felineus* metacercariae. New data on the presence and localisation of the components of the serotonergic system in metacercariae of *O. felineus* were obtained confirming the important role of the serotonin signalling system studied in activity of the vital functions of the worms.

### 4.2. Serotonin and 5-HT_7_-IR in Cysticercoid Larvae of Hymenolepis diminuta

Serotonin signalling system components were earlier identified in the cestode *Hymenolepis diminuta.* Thus, serotonin was detected in larvae and adults of *H. diminuta* [25,36,128,129]. The localisation and distribution of serotonergic neurons and neurites in the cysticercoid of *H. diminuta* nervous system was determined and described in the main nerve commissure, the lateral and rostellar ganglia, the longitudinal nerve cords and their connectives. In the peripheral nervous system, 5-HT-IR nerve fibres occur in the nerve plexus within each sucker [131].

Moreover, information on serotonergic elements in *H. diminuta* was revealed in several earlier studies. Thus, the serotonin synthesis pathway was identified. The key enzyme for serotonin synthesis—the 5-hydroxytryptophan hydroxylase—was detected [25,37,47,48,49]. It was shown that *H. diminuta* are capable not only of synthesising but actively absorbing the serotonin from the host, having a specialised transport system [38,132]. The functional roles of serotonin were also studied. Thus, in *H. diminuta* serotonin induced contractions of the muscle strip preparations [53], influenced the carbohydrate metabolism [133], migration in the host organism [134,135] and parasite reproduction [136].

Our data on the presence of serotonergic structures in the central and peripheral compartments of the nervous systems of cysticercoids of *H. diminuta* confirmed the previously published findings and indicates the importance of the neurotransmitter in worm physiology. In our study, morphometric measurements were performed and new morphological parameters of serotonergic nerve components were estimated in cysticercoid of *H. diminuta* nervous system.

#### 4.2.1. Musculature and Nerve Fibres

Immunoreactivity to 5-HT_7_ was also found on/along the muscle elements of worm body walls and suckers. Considering the important role of serotonin in worm motor activity, it can be assumed that 5-HT_7_ type receptor can mediate the serotonin action in the cysticercoids. Our study reveals a weak immunostaining to 5-HT_7_ in longitudinal nerve cords and nerve commissure of *H. diminuta* larvae. Little is known about the functional properties of the serotonin receptors in cestodes. The available data are restricted to the recent identification of three types of serotonin receptors in *E. granulosus* and *M. corti* [82]. Using several approaches, these have been identified as G protein-coupled proteins, exhibiting a strong sensitivity to serotonin and unique characteristics in differential responsiveness to ligands. The localisation studies performed with a fluorescent probe showed a punctiform staining pattern concentrated in the suckers of *E*. *granulosus* and *M. corti* larvae [82].

#### 4.2.2. The Flame Cells

We observed 5-HT_7_-IR in cysticercoid flame cells of *H. diminuta*. Flame cells are ciliated cells located within the basal matrix, at the cestode neodermal tissue [137]. They are considered as terminal cells in bulbs [138] which are the basic units of the protonephridial (or excretory) system of invertebrates, including flatworms. The excretory system plays an important role in parasite physiology. They need to maintain their tissue osmotic pressure against that of the host environment [139]. The excretory system of parasitic worm acts like osmoconformer, facilitating the conservation of water and eliminating of salts to survive in the intestine or body cavities of their hosts [139]. According to published data, the structural characteristics of which were described, a filamentous actin (F-actin) was found in the flame cells of cestodes *T. solium*, *Diphyllobothrium dendriticum*, *H. diminuta*, monogenean *Gyrodactylus rysavyi* [137,140,141,142] and trematode *S. mansoni* [143]. The presence of F-actin could be related to their contractile motions [140,141]. In our study, the phalloidin staining confirmed the presence of actin myofilaments in the flame cells of the cysticercoid of *H. diminuta,* which corresponds to the data obtained for other cestode species.

Our data on the presence of 5-HT_7_ immunoreactivity in the flame cells of the cysticercoid larvae of *H. diminuta* indicates that the 5-HT_7_ receptor may be involved in the regulation of the contractile activity of the flame cells. More research is needed to elucidate 5-HT and 5-HT_7_ signalling in the functioning of the excretory system in flatworms.

Thus, the distribution of 5-HT_7_-IR was, for the first time, studied in larvae of the cestode *H. diminuta* and the trematode *O. felineus*. The immunocytochemical results, together with some bioinformatics reviews, show the presence of a specific type of 5-HT_7_ serotonin receptor in the trematodes and cestodes. It can be assumed that the serotonin 5-HT_7_ type receptor can provide the implementation of the mechanisms of serotonin action in the flatworms studied. However, for justification of the serotonin receptor activation in the studied species, further research is required. The future research on molecular characterisation (cloning and sequencing) of the serotoninergic receptors in worms studied will provide the novel data on gene structure and facilitate further pharmacological investigation.

Flatworms occupy a key position in animal evolution in which cephalisation and an organised nervous system first appeared. Therefore, the study of this group of animals plays an important role in determining early nervous system development, the evolution of the nervous system and neuronal signalling pathways. Our data on the presence and localisation of serotonergic components in flatworms serve as a foundation for the better understanding of serotonin signalling in their organisms. A comparative approach to the study of the functioning of neurotransmitter systems in the parasites and their hosts contributes to the solution of a fundamental scientific problem associated with the complex of evolutionary fixed mechanisms of host-parasite interactions. Taking into account the important roles of biogenic amine, serotonin, in parasitic worms, the serotonergic compartments of the nervous system could also be considered as potential targets for anti-parasite drugs.

## 5. Conclusions

The distribution of immunoreactivity to the 5-HT_7_ serotonin receptor was investigated in larvae tissues of two parasitic flatworms, the trematode *Opisthorchis felineus* and the cestode *Hymenolepis diminuta* for the first time.The presence of the specific serotonin 5-HT_7_receptor’s immunoreactivity in the studied parasitic worms has been shown. It emphasises the importance of the serotonergic signalling system for realisation of vital functions in representatives of Platyhelminthes.The results suggest that the 5-HT_7_ type of serotonin receptor can mediate the serotonin action in the studied species and is an important component of worm motor system control.Taking into account the important roles of 5-HT in parasite biology, the present report also suggests that the flatworm serotoninergic nervous system could be considered a target for anti-parasite drugs.

## Figures and Tables

**Figure 1 biomolecules-11-01212-f001:**
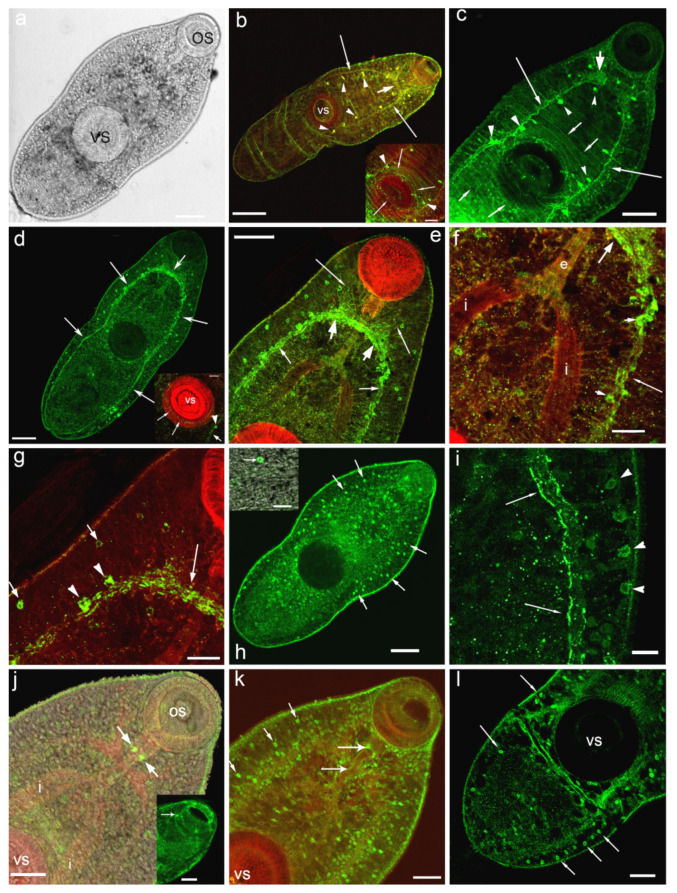
Immunoreactivity to serotonin (**b**,**c**) and 5-HT_7_ serotonin receptor (**d**–**l**) in metacercariae larvae of *Opisthorchis felineus* (in green) and histochemical staining of muscles with phalloidin (in red). (**a**)—overview of metacercariae of *O. felineus*, transmission light microscopy, whole mount; (**b**)—5-HT-IR (green) in the central nervous system—the brain ganglia and brain commissure (thick arrow), in the ventral nerve cord (long thin arrows), the serotonergic neurons of the nerve cords are indicated by arrowheads, *inset*: the 5-HT-IR cells (arrowheads) and fibres innervating the ventral sucker (thin arrows), running from the ventral nerve cords; TRITC-phalloidin staining for muscles are shown in red; (**c**)—anti-serotonin staining in cephalic ganglion (thick arrow), ventral nerve cords (long arrows), in transversal commissures connecting the nerve cords (short arrows), serotonergic neurons in head ganglion and in the nerve cords are indicated by arrowheads; (**d**)—anti-serotonin receptor 5-HT_7_-IR (green) in the longitudinal nerve cords (long thin arrow), in the brain ganglia and brain commissure (short arrow), inset: 5-HT_7_-IR in the ventral sucker (long arrows), in the nerve cord at the level of ventral sucker (short arrow) and in the round body (arrowhead); (**e**)—5-HT_7_-IR in brain ganglia and brain commissure (thick arrows), in the major nerve cords (thin short arrows) and in the anterior nerves (long arrows); the round structures scattered through the body are also visible; (**f**)—a higher magnification, 5-HTR_7_-IR in the brain ganglion (thick arrow) and in the lateral nerve cords (thin long arrows), in neurons of the nerve cord (short arrows); (**g**)—5-HT_7_-IR in brain commissure (long arrow), in the neurons within the major nerve cords (arrowheads) and in the round structures, probably the gland cells or the other unidentified cells along the lateral body side (short arrows); (**h**)—5-HT_7_-IR in the round structures (the secretor gland cells or the other unidentified cells) scattered throughout the worm body (long arrows), inset: the larger magnification of the round structure (arrow); (**i**)—5-HTR_7_ along the major nerve cords (arrows) and in the round structures (arrowheads), the posterior body region; (**j**)—5-HT_7_-IR in two serotonergic neurons located near pharynx (arrows), inset: 5-HT_7_-IR in the structures, located in the structures (the sensory papillae) located in the oral sucker (arrow); (**k**)—5-HT_7_-IR in serotonergic neurons and fibres located near pharynx and esophagus (long arrows), the round structures scattered throughout the worm body are visible (short arrows); (**l**)—a row of 5-HT_7_ immunopositive structures in posterior region of the worm (long arrows) and near the ventral sucker. Scale bars: (**a**,**c**–**e**,**h**)—50 μm; (**b**)—100 μm; (**g**,**f**), inset on (**d**)—20 μm; (**i**), inset on (**h**)—10 μm, (**j**–**l**); inset on (**g**)—30 μm. Abbreviations: e—esophagus, i—intestine branches, os—oral sucker, vs—ventral sucker.

**Figure 2 biomolecules-11-01212-f002:**
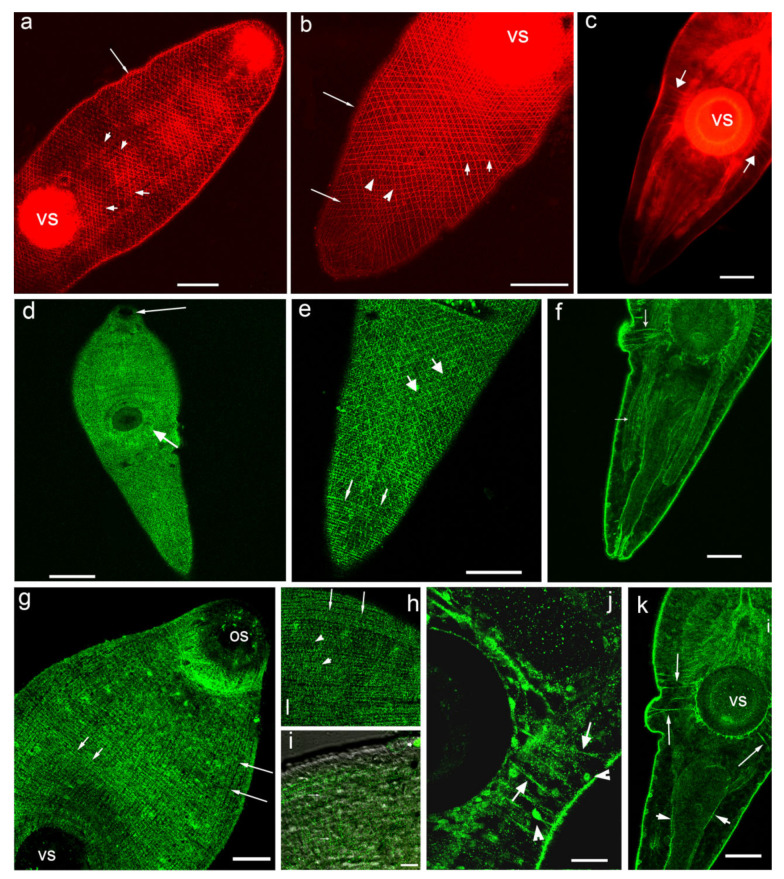
Histochemical staining of the body musculature with phalloidin (red, (**a**–**c**)) and immunoreactivity to 5-HT_7_ serotonin receptor (green, (**d**–**k**)) in metacercariae larvae of *Opisthorchis felineus. (***a**,**b**)—phalloidin staining of body wall musculature in the anterior (**a**) and posterior (**b**) body parts of *O. felineus*, the longitudinal (long arrows), diagonal (short arrows) and circular (arrowheads) muscle fibres, vs—the ventral sucker; (**c**)—muscle fibres extending from the body wall to the ventral sucker (short thick arrows), vs— ventral sucker; (**d**)—staining to serotonin receptor (5-HT_7_-IR) in the body wall muscles of *O. felineus*, overview, os— the oral sucker (long arrow), vs—the ventral sucker (short arrow); (**e**)—posterior body region, the pattern of 5-HT_7_-IR shows the circular (thin arrows) and diagonal (thick arrows) muscle fibres; (**f**)—5-HT_7_-IR in the muscle fibres extending from the body wall to the ventral sucker and in the longitudinal muscles of the intestine (arrows); (**g**)—the body wall musculature of the anterior region of the body, stained with 5-HT_7_-IR, the longitudinal (long arrows) and circular (short arrows) filaments are indicated, part of the oral sucker musculature (os) is visible; (**h**,**i**)—more detailed views on the 5-HT_7_-IR elements along the body muscles: (**h**)—the longitudinal (long arrows) and diagonal (short arrows) muscle fibres are indicated; (**i**)—punctate pattern of 5-HT_7_-IR along the circular muscle fibres (arrows), confocal and transmission light microscopy; (**j**)—5-HT_7_ immunoreactivity in muscle fibres running from the body wall to the ventral sucker (arrows) and in the round structures along the body margin (arrowheads); (**k**)—5-HT_7_-IR in the posterior body region, the muscle fibres running from the body wall to the ventral sucker (long arrows) and the longitudinal muscles of the excretory bladder (short arrows), vs—ventral sucker. Scale bars: (**a**–**c**,**e**,**f**,**k**)—50 μm; (**d**)—100 μm; (**g**)—30 μm; (h), (**i**)*—*10 μm; (**j**)—20 μm.

**Figure 3 biomolecules-11-01212-f003:**
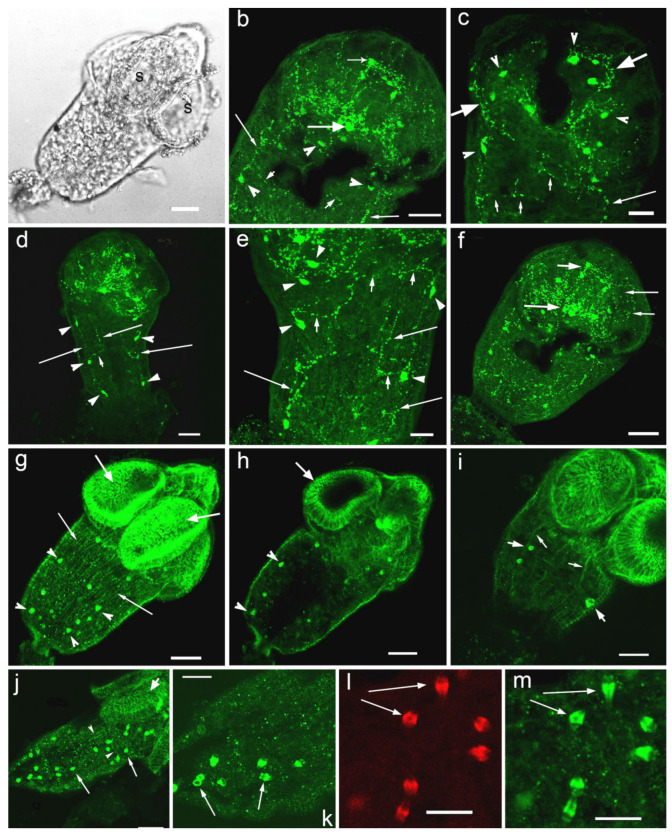
Immunoreactivity to serotonin (**b**–**f**) and 5-HT_7_ serotonin receptor (**g**–**l**) in cysticercoid larvae of *Hymenolepis diminuta.* (**a**)—cysticercoid of *H. diminuta* with suckers (s) in the scolex, transmission light microscopy, whole mount; (**b**)—5-HT-IR in the scolex of *H. diminuta*, the lateral ganglia (thick arrow) and the rostellar ganglia (short arrow), the serotonergic neurons (arrowheads), the ventral nerve cords (thin long arrows) and transversal commissures between them (thin short arrows), the serotonergic nerve fibres (the rostellar nerves) extending from the lateral to the rostellar ganglia are visible; (**c**)—5-HT-IR in the longitudinal nerve cords (long thin arrow), in commissures between them (short arrows), in the nerve cells along the nerve cords (arrowheads), in the nerve plexus and initial part of the longitudinal nerve cords (short thick arrows); (**d**)—several pairs of 5-HT-IR neurons (arrowheads) situated along the major nerve cords (long thin arrows), the lateral ganglia are visible; (**e**)—5-HT-IR in the longitudinal nerve cords (thin long arrows) and in the commissures between them (short arrows), in the nerve cells along nervous cords (arrowheads); (**f**)—5-HT-IR in the lateral ganglion (long thick arrow), in the rostellar ganglion (short thick arrow), in the nerve fibres of the plexus within suckers (thin arrows); (**g**)—5-HT_7_-IR along the longitudinal and transversal muscle fibers comprising the worm body wall (thin long arrows) and in the muscles of suckers (thick arrows), 5-HT_7_-IR is visible in the oval structures, probably the flame cells of the excretory system (arrowheads); (**h**)—5-HT_7_-IR along the radial the muscle of the suckers (short arrow) and in the oval structures, the flame cells of the excretory system (arrowheads); (**i**)—5-HT_7_-IR in the longitudinal muscle fibres of body (thin arrows), the 5 HT_7_ -IR is visible in the oval structures (short thick arrows), the flame cells of the excretory system; (**j**)—5-HT_7_-IR in the longitudinal and transversal (arrowheads) muscle fibres of the body and in the muscles of the sucker (short thick arrow), 5-HTR_7_-IR is also present in the oval structures, the flame cells of the excretory system (long arrows); (**k**)—5-HT_7_-IR in the oval structures (long arrows), the flame cells, scattered in the cysticercoid body; (**l**)—phalloidin staining of the oval structures having positive 5-HT_7_-IR, the larger magnification; (**m**)—5-HT_7_-IR in the oval structures (long arrows), presumably, the flame cells of the excretory system, the large magnification. Scale bars: (**a**,**b**,**d**,**f**,**g**–**j**)—20 μm; (**c**,**e**,**k**–**m**)—10 μm.

**Figure 4 biomolecules-11-01212-f004:**
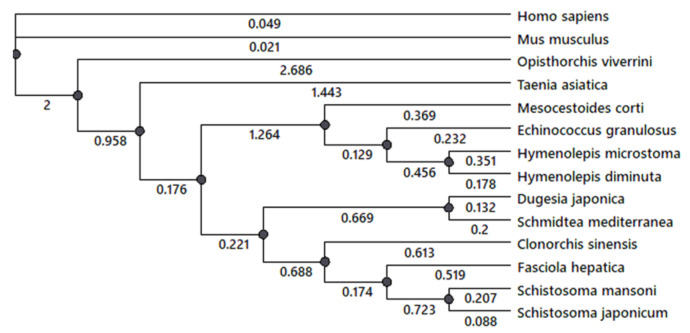
A phylogenetic tree of serotonin 5-HT_7_ receptors in cestodes, trematodes and planarian species with the annotated genome sequences.

## Data Availability

Not applicable.

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
