# Peer review of "Serotonin Signalling in Flatworms: An Immunocytochemical Localisation of 5-HT7 Type of Serotonin Receptors in Opisthorchis felineus and Hymenolepis diminuta"

_biomolecules, 2021, doi:10.3390/biom11081212_

Round 1

Reviewer 1 Report

This is an interesting study describing the distribution of serotonin and serotonin receptor (5-HT7) in Opisthorchis felineus and Hymenolepis diminuta through use of immunocytochemical and microscopy methods.  The authors provide a very detailed description of the spatial distribution of 5-HT and 5-HT7, and highlight that serotonin receptors are localised to muscle which aligns with a role in muscular function.  This study provides interesting data for especially for researchers focussed on the serotoninergic pathway in flatworms.  The manuscript is detailed, broadly covers key literature, and the image presentation in the figures is excellent.  This work is important as it lays foundations to understanding of 5-HT signalling in flatworms, especially in the broader context of receptor evolution.

The authors employ mainly immunolocalisation techniques to generate the data presented.  The immunocytochemistry experiments appear to be robust and provide informative data.  The authors employ commercially available primary antibodies raised against the 5-HT7 receptor – it would be useful to provide information on the cross-reactivity of this antibody with the 5-HT7 receptor sequence in O. felineus and H. diminuta – was the antibody raised against an epitope that is conserved in the flatworm receptors?  It is not clear if the authors cloned and sequenced the 5-HT7 receptors in O. felineus and H. diminuta, or predicted the sequences via bioinformatic approaches.  More clarity here is needed.  How did the authors ensure the specificity of the staining achieved.

Overall, this paper presents useful data that should be communicated to researchers working in this area.  In my opinion it would be suitable for publication in Biomolecules, and represents information that has the potential to contribute to further progress in understanding classical transmitter signalling in flatworms.  However the manuscript requires significant language editing to ensure understanding of content and key outcomes.  Care should be taken with the language used - for example the authors have not identified serotonin (5-HT) and serotonin receptor (5-HT7) in the flatworm species but rather have localised these receptors.  The manuscript would benefit from the molecular characterisation (cloning and sequencing) of the serotoninergic receptors that would provide novel data on gene structure, and facilitate further pharmacological investigation.  In addition bioinformatic investigation to mine for other 5-HT receptor subtypes in the species of interest would also be informative - especially in the context of cross-reactivity of the antibody employed.

Reviewer 2 Report

The paper by Kreshchenko et al is a well-executed experiment that provides definitive evidence
that the 5-HT7 receptor is present in Opisthor- 3 chis felineus and Hymenolepis diminuta. While
the paper is strong, there are a few concerns that I have. They are described below. Please
make the necessary modifications.
1] some of the language in the abstract is troublesome. For example, the sentence “5-HT7-IR
was evident in the 19 round bodies, probably the glandular cells scattered in the larva body”.
Why is it probably? Do you not know? If you do not know, don’t speculate. I’d suggest editing
the abstract heavily to better describe the paper.
2] typo on line 73 “reman poorly studied” should be remain
3] methods section is small. More detail on procedure is required.
4] there is a lot of repetition throughout the paper. The discussion opens with a near identical
statement that was made in the Introduction. Please correct this.
5] your figures are phenomenal
6]] While I appreciate the significance or studying all aspects of science and biology, the paper
does not discuss the significant of these results. Yes, serotonin is important in humans, and they
were observed in these parasitic worms, but what does this show us beyond “there’s serotonin
present”? Additional info on the implications and significance of the data would be beneficial.

Author Response

Suggestions by Reviewer2:

1] some of the language in the abstract is troublesome. For example, the sentence “5-HT7-IR
was evident in the 19 round bodies, probably the glandular cells scattered in the larva body”.
Why is it probably? Do you not know? If you do not know, don’t speculate. I’d suggest editing the abstract heavily to better describe the paper. = done
2] typo on line 73 “reman poorly studied” should be remain= done
3] methods section is small. More detail on procedure is required.==added
4] there is a lot of repetition throughout the paper. The discussion opens with a near identical
statement that was made in the Introduction. Please correct this. =done

5] your figures are phenomenal= Thank you!
6]] While I appreciate the significance or studying all aspects of science and biology, the paper does not discuss the significant of these results. Yes, serotonin is important in humans, and they were observed in these parasitic worms, but what does this show us beyond “there’s serotonin present”? Additional info on the implications and significance of the data would be beneficial. = done

Dear colleague, we are thankful so much for you reviewing our MS and your valuable suggestions helping us to improve our MS!

We would like to say that originally the paper was intending to be a review on serotonin receptors identification in flatworms and then we decided to add some our experimentally data we have on two worms species (for discussion this matter), this part is not very big that’s why the methods section is also small.

We performed all necessary modification suggested by reviewer 2:

  1. We checked and carefully corrected all English mistakes and typos across the MS (alone, and with English editing service as well).
  2. The Abstract was corrected according the suggestions; we removed all uncertences from there.
  3. Some details were added to the methods section, subdivided it on paragraphs.
  4. We removed some repetitions from the discussion section and corrected it.
  5. Additional essential statement about biological significance of the study was added at the end the text:

Flatworms occupy a key position in animal evolution when the cephalization and an organized nervous system were first appeared. Therefore, the study of this group of animals plays an important role in determination of the early development of the nervous system, the evolution of the nervous system and the neuronal signaling pathways. Our data on the presence and localization of serotonergic components in flatworms are serving as a foundation for better understanding of serotonin signaling in their organism. A comparative approach to the study the functioning of neurotransmitter systems in the parasites and its hosts contributes to the solution of a fundamental scientific problem associated with the complex of evolutionarily fixed mechanisms of host-parasite interactions. Taking into account the important roles of biogenic amine, serotonin in parasitic worms, the serotonergic compartments of the nervous system could also be considered as potential target for anti-parasite drugs.

  1. We also performed some changes in the text of MS according to other reviewer’s suggestion.

Sincerely, Natalia Kreshchenko
